# End user experiences of an electronic health records platform in a tertiary hospital system in Kenya

Anmol Shrestha[1☯], Jasmit Shah[2,3☯], Zamanali Khakhar[4], Armando Ruiz[1], Zohray Talib[1], Sayed K. Ali[2]*

**1** California University of Science and Medicine, Colton, California, United States of America,
**2** Department of Medicine, Aga Khan University, Nairobi, Kenya, **3** Brain and Mind Institute, Aga Khan University, Nairobi, Kenya, **4** School of Medicine, University of Nairobi, Nairobi, Kenya

☯ These authors contributed equally to this work.
* sayed.karar@aku.edu

## Abstract

### Background

Electronic Health Record (EHR) systems allow health care facilities to provide better care to patients and improve overall provider efficiency. They are vital for low-and middle-income countries in achieving the United Nations Sustainable Development Goal of ensuring healthy lives and promoting the well-being of all their citizens. This study aimed to evaluate perceptions and usage of a comprehensive EHR by end-users after its deployment into a tertiary hospital system and its outpatient centers in Kenya and also aimed to understand the effectiveness of various implementation strategies deployed by the institution during their implementation process.

### Methods

We conducted a cross-sectional study between October 2023 and January 2024 at the Aga Khan University Hospital, Nairobi with staff involved in using the EHR system. A standardized electronic questionnaire was shared with the users and responses were captured on REDCap after obtaining written consent electronically.

### Results

Of the 548 participants who agreed to join the study, 471 responded to the survey in full. Most respondents (420, 89.9%) stated that the EHR made their work better or much better, compared to a paper-based system. Majority of end-users stated that the EHR benefited their practice (378, 87.1%), provided autonomy to healthcare workers (382, 86.2%), the quality of healthcare (424, 95.5%), interactions within the healthcare team (353, 79.1%) and enjoyment of their clinical practice (360, 80.7%).

**Data availability statement:** All relevant data are within the paper and its Supporting information files.

**Funding:** The author(s) received no specific funding for this work.

**Competing interests:** The authors have declared that no competing interests exist.

## Conclusion

The majority of end users believed the EHR to be effective and appropriate to use within this specific and unique healthcare system. The specific strategies deployed by institution were also successful in ensuring high rates of EHR usage and can be looked at as a blueprint for future EHR deployments in other sub-Saharan African healthcare systems.

---

## Introduction

Electronic Health Records (EHRs) have long been utilized by high income countries (HICs) for streaming patient documentation, reporting, billing communication between health care workers (HCWs) in patient care [1]. They allow HCWs to make care decisions with the most up-to-date clinical information and facilitate high quality health care by detailing critical information regarding patients' conditions and current medications to ensure patients are not given contraindicated treatment regiments [2].

EHR's are essential for low- and middle-income countries (LMICs) aiming to optimize their health system through the Quadruple Aim model of improvements in population health, patient experience and provider experience as well as reduction in overall costs [3]. EHRs' efficient reporting systems not only improve patient health directly; they also improve population health by enabling health systems to more easily respond to the health challenges of the 21st century, such as the double burden of diseases from non-communicable and communicable diseases [4,5]. They additionally improve patient experience through a documentation process that removes the physical burdens of paper-based documentation [6]. Finally, the transition to an EHR reduces the health care costs associated with the purchasing and storage of paper-based records and addresses provider burnout through a streamlined and up-to-date documentation interface that minimizes administrative work [1,2].

### Types of EHR deployments in LMICs

As such, the successful implementation and effectiveness of EHRs in improving patient care are important areas of study to achieve health equity such as equitable access to healthcare services, and reduction of disparities in treatment quality for LMICs. There are no standardized guidelines or terminology in how LMICs should deploy EHRs and as a result, their integration into health systems vary from country to country. Ethiopia and Malawi deployed EHRs specifically into their central referral tertiary hospitals [7–9]. Their EHRs were created with assistance and support from HIC institutions and were deployed throughout the hospitals, from the wards to laboratories to pharmacies. End users' surveys in these hospital systems stated that the efficiency of the EHR was useful in tertiary hospitals and their high patient volume, especially when compared to paper-based systems. End users also stated that the inter departmental operability of the EHR allowed them to coordinate complex patient care. These deployments, however, failed to create a sustainable impact due to a lack of training and poor technical support [7–9]. Similar findings were found

in Ghana, where EHRs were deployed across several tertiary hospitals. Mensah et al conducted a series of qualitative interviews in these hospitals and found that the most common complaints for poor EHR integration were failing infrastructure and lack of administrative support, in addition to poor training and technical support [10]. These hurdles were the most commonly shared sentiments attributed by HCWs as the main reasons for EHR deployment failures across several other health systems [11–13]. These surveys also simply looked at whether EHRs were efficacious in their deployments into singular hospitals. There have been no studies, to the best of our knowledge, which have surveyed end users on the effectiveness of an EHR deployment in a sub-Saharan Africa tertiary hospital system on improving clinical care.

**EHR deployment in Kenya**

In 2022, the Aga Khan University Hospital in Nairobi (AKUHN), Kenya began deployment of, a comprehensive EHR developed by Medical Information Technology INC, a private software firm from the United States [14]. AKUHN is a 300-bed tertiary center with major departments that serve patients throughout Kenya and neighboring countries. It is also the coordinating body for over 50 outreach centers throughout Kenya, each offering a range of healthcare services.

The EHR system, along with the necessary infrastructure, was implemented across multiple departments at the main hospital— including billing, pharmacy, registration, and all clinical areas— as well as at all outreach centers. The hospital leaned on previous research of end-user experiences to develop their deployment plan [14]. To support the rollout, AKUHN established governance structures to enhance communication among leadership, the Information and Communication Technology (ICT) team, the clinical informatics team, and end users. A dedicated clinical informatics team – composed of in-house physicians and nurses was formed. Team members were assigned specific roles based on their prior clinical experiences, underwent extensive training on the EHR, and subsequently trained other staff. End-user training (live training as well as video tutorials) began two months prior to "Go-Live" date to ensure user readiness and comfortability with the system. The clinical informatics team also provided vital frontline support during and after "Go-Live". Their responsibilities included aligning EHR workflows with clinical practices, troubleshooting issues in real-time, reinforcing training, and acting as a bridge between clinical and technical teams to ensure safe and efficient patient care. The ICT team included a diverse range of specialists essential to the successful deployment of the EHR system. The Business Process Team focused on aligning the EHR system with existing clinical and administrative workflows to ensure the system was efficient, compliant and user-centered. Software developers handled customization, integration, and technical support to ensure the EHR system functioned seamlessly within the organization's digital environment while the network engineers ensured stable connectivity and server performance for real-time access across all locations. During the "Go-Live" phase, the ICT team played a pivotal on the ground role; resolving login and access issues, ensuring appropriate permissions and secure access, and monitoring system performance to quickly address any disruptions. Their combined efforts were essential in minimizing downtime, supporting end-users, and ensuring a smooth transition to the new system.

To maintain transparency and foster buy-in, the administration organized monthly town hall meetings starting six months prior "Go-Live". These were conducted live and online, offering stakeholders updates on implementation milestones, addressing questions and gathering feedback. These sessions were key in engaging clinical staff – considered critical stakeholders in the success of the EHR rollout. Following Go-Live, both technical and clinical support remained available onsite to assist end users in adapting to the new system and to enhance the overall implementation experience. Through this comprehensive deployment strategy, we aim to understand not only whether the EHR improved patient care, but also the experiences and perceptions of the clinical staff who interacted with the system.

**Methods**

We conducted a cross-sectional study between 1st October 2023 and 31st January 2024, 11 months after successful deployment of the EHR system. This study involved all clinical staff at the Aga Khan University Hospital, Nairobi involved

in the implementation and use of the EHR system. The study targeted all users of the EHR including doctors of different cadres (consultants, fellows, residents and other medical officers/learners), all clinical nurses involved in patient care and allied health staff (pharmacy staff, radiology staff, laboratory staff, physiotherapists, occupational therapists, and nutritionists).

Email addresses and/or WhatsApp contact details of all eligible staff were obtained by the principal investigator with the help of the Information Technology Department. Email or WhatsApp invitations with a link to a voluntary, deidentified survey was sent to all eligible staff. Responses from the participants remained anonymous. Online survey data were collected through the Research Electronic Data Capture (REDCap) platform (Vanderbilt and National Institute of Health) [15]. The informed consent process was conducted through REDCap and participants were presented with a detailed consent form, which they reviewed and electronically signed before proceeding. The consent procedure, including the use of electronic consent, was reviewed and approved by the University Ethics Committee. A modified standardized questionnaire adapted from the Michigan Public Health Institute EHR End-User survey was used in this study [16]. This contained questions evaluating different domains including the EHR use environment, EHR impact, and EHR functionality. No personal identifiers were collected from participants. Categorical data were analyzed as frequencies and percentages. Fisher's exact test was used to compare gender and healthcare differences. For multiple comparisons in each section, the p value was adjusted using the Bonferroni method. Data analysis was performed using SPSS statistical software V.20.0 (IBM). The significance level was set at $\alpha = 0.05$, and all tests were two tailed. This study was approved by the Institutional Scientific Ethics and Review Committee (ISERC) at the Aga Khan University (2023/ISERC-63 (v3)) and National Commission for Science, Technology and Innovation (NACOSTI) (NACOSTI/P/23/29466).

## Results

### Participant characteristics

A total of 554 participants (all end-user with access to the EHR) were contacted about participating in the study with 548 (99%) agreeing to join the study. Of the 548, 85.9% (n = 471) responded to the survey in full. More than half, 57.7% were females (272/471) and 82.3% were of age 30–49 years. Majority of the participants were nurses at 47.2% (n = 220), followed by doctors at 26.6% (n = 124) and allied staff at 26.2% (n = 122). In terms of physician roles, 44.7% were consultants, 23.6% were fellows and residents and 22.0% were medical officers/learners. Table 1 below describes the overall characteristics of participants.

### Experience with EHR

The majority of respondents, 89.4% (n = 421) stated that they have been using the EHR system for>= 6 months. Only 21.7% (n = 102) stated that they had used an EHR prior to using the EHR at AKUHN and 86.3% (n = 88) rated the current EHR as being better or much better than their previous EHR experience. Furthermore, 89.9% (n = 420) stated that the use of the EHR has made their work better or much better compared to a paper-based medical record system Table 2 describes the overall experience with EHR.

### EHR use environment

The majority used desktop computers (90.2%), followed by laptops (29.7%). After the full deployment of the EHR, respondents rarely used paper records. Only 8.7% stated using paper records to see tests and results, 8.8% stated using paper records to obtain contact information for patients, and 10.9% to document something in the patient medical records.

Based on the EHR use environment, majority of the respondents had agreed that they received adequate training on EHR usage (95.9%, n = 441), felt their questions about the EHR were sufficiently answered (94.3%, n = 434), felt they received technical support whenever they needed it (92.8%, n = 427), and were satisfied with the technical support they

**Table 1. Overall characteristics of study participants.**

| Gender | Male | 199 | 42.3% |
|---|---|---|---|
| | Female | 272 | 57.7% |
| Type of Healthcare Category | Allied Health Staff | 122 | 26.2% |
| | Nurse | 220 | 47.2% |
| | Doctor | 124 | 26.6% |
| Specify the Doctor | Consultant/ Instructor | 55 | 44.7% |
| | Resident | 29 | 23.6% |
| | Fellow | 10 | 8.1% |
| | House/ Medical Officer | 27 | 22.0% |
| | Medical Student | 2 | 1.6% |
| Doctors Primary Department | Anesthesiology/ Critical Care | 8 | 6.7% |
| | Pathology | 10 | 8.4% |
| | Emergency Medicine | 16 | 13.4% |
| | Family Medicine | 10 | 8.4% |
| | Hematology, Oncology and Palliative Care | 5 | 4.2% |
| | Imaging and Diagnostic Radiology | 7 | 5.9% |
| | Internal Medicine/ Critical Care | 31 | 26.1% |
| | Obstetrics and Gynecology | 6 | 5.0% |
| | Paediatrics and Child Health | 18 | 15.1% |
| | Surgery | 8 | 6.7% |
| Current Age (years) | < 30 years | 29 | 6.2% |
| | 30 - 39 years | 215 | 45.8% |
| | 40 - 49 years | 171 | 36.5% |
| | 50 - 59 years | 52 | 11.1% |
| | >= 60 years | 2 | 0.4% |

**Table 2. Overall experience of EHR study participants.**

| EXPERIENCE WITH EHR | | n | % |
|---|---|---|---|
| How long have you been using the EHR at AKUHN (months) | < 6 months | 50 | 10.6% |
| | >= 6 months | 421 | 89.4% |
| How has the use of the EHR changed your work compared to using a paper-based medical record system? | Much Worse/ Worse | 20 | 4.3% |
| | Neither worse nor better | 27 | 5.8% |
| | Better/ Much Better | 420 | 89.9% |
| Prior to using this EHR at AKUHN, have you used another EHR system? | Yes | 102 | 21.7% |
| | No | 367 | 78.3% |
| How would you rate this EHR at AKUHN as compared to other EHR systems you used in the past? | Much Worse/ Worse | 5 | 4.9% |
| | Neither worse nor better | 9 | 8.8% |
| | Better/ Much Better | 88 | 86.3% |

received (89.6%, n = 414). In terms of EHR failure, 55.4% (n = 255) respondents felt the system downtimes were acceptable, whereas only 27.2% (n = 125) felt that there were policies and procedures to allow clinicians to continue patients when the EHR was down. Less than half of the participants (42.4% and 40.8%) felt that laboratory and radiology results appeared in the EHR in a timely manner. The majority responded that there were adequate computer terminals to access the EHR (74.7%) and that the EHR responded quickly to their actions (68.2%). The respondents felt that during the

implementation process, the plans were communicated adequately (84.5%) and that adequate resources were committed to the implementation process (86.2%). Table 3 summarizes the overall statistics of the EHR use environment.

## EHR impact and functionality

Overall, respondents had very positive views of the EHR. They stated that the EHR benefited their practice (87.1%), provided autonomy to HCW (86.2%), the quality of healthcare (95.5%), interactions within the healthcare team (79.1%) and enjoyment of their clinical practice (80.7%). The respondents also believed that the EHR benefited stress levels (74.9%), healthcare providers self-image (76.2%), humaneness of clinical practice (74.0%), the report between providers and patients (76.9%), personal and professional privacy (85.5%), keeping the provider up-to-date on patient information (91.1%), patient satisfaction (87.6%), generalist ability to manage more complex problems (87.2%), comprehensiveness of patient care (91.1%), clinical efficiency (92.4%) and avoiding errors (89.0%).

The respondents also found the EHR highly user friendly. They rated the EHR was "Easy" to use when obtaining and reviewing patient data (96.8%), documenting care for my patients (97.9%), viewing lab results (98.2%), viewing radiology results (97.5%), preventing adverse events (95.8%), tracking patients' preventive care (94.2%), managing chronic disease conditions (94.5%), managing orders (96.5%), managing referrals (96.3%), providing patients education materials (92.0%), analyzing outcomes of care (95.3%), accessing the EHR from offsite locations (97.9%), coordinating patient care (93.0%), communicating with patients (91.4%), and enhancing continuity of care (95.8%). They felt that there were no missing functionalities in the EHR (92.8%) and did not find that they needed additional help when using the EHR (95.5%). Table 4 summarises the EHR impact and the functionality.

## Overall evaluation of the EHR

Respondents agreed the EHR was easy to use (93.8%) and the screens were user friendly (92.3%). The EHR was also found to have all the functionalities the respondents expected (89.9%) and overall they were satisfied with their

**Table 3. EHR use environment of study participants.**

| EHR USE ENVIRONMENT | Disagree | | NOR | | Agree | |
|---|---|---|---|---|---|---|
| I received adequate training on how to use this EHR | 7 | 1.5% | 12 | 2.6% | 441 | 95.9% |
| My questions about use of this EHR were sufficiently answered | 5 | 1.1% | 21 | 4.6% | 434 | 94.3% |
| I receive technical support whenever I need it | 10 | 2.2% | 23 | 5.0% | 427 | 92.8% |
| I am satisfied with the technical support I have received in use of this EHR | 10 | 2.2% | 38 | 8.2% | 414 | 89.6% |
| The system downtimes are acceptable | 28 | 6.1% | 177 | 38.5% | 255 | 55.4% |
| When this EHR system is down, we have policies and procedures to allow the clinician to continue to see patients | 254 | 55.2% | 81 | 17.6% | 125 | 27.2% |
| The EHR screens respond to my actions instantly | 37 | 8.0% | 110 | 23.8% | 315 | 68.2% |
| Our facility has adequate computer terminals to access this EHR | 35 | 7.6% | 81 | 17.6% | 343 | 74.7% |
| Lab results appear in this EHR in a timely fashion | 103 | 23.1% | 154 | 34.5% | 189 | 42.4% |
| Radiology results appear in this EHR in a timely fashion | 110 | 24.6% | 155 | 34.6% | 183 | 40.8% |
| This EHR allows me to review trends in lab values | 23 | 5.2% | 51 | 11.4% | 372 | 83.4% |
| The project plan was adequately communicated to us during implementation | 15 | 3.3% | 55 | 12.1% | 383 | 84.5% |
| Adequate resources were committed to the implementation | 14 | 3.1% | 48 | 10.7% | 387 | 86.2% |

**Table 4. EHR Impact and Functionality of study participants.**

| EHR IMPACT | Detrimental | | NOR | | Benefit | |
|---|---|---|---|---|---|---|
| Costs to my practice in providing care | 26 | 6.0% | 30 | 6.9% | 378 | 87.1% |
| Autonomy of healthcare providers | 16 | 3.6% | 45 | 10.2% | 382 | 86.2% |
| Quality of health care | 2 | 0.5% | 18 | 4.1% | 424 | 95.5% |
| Interactions within the healthcare team | 33 | 7.4% | 60 | 13.5% | 353 | 79.1% |
| Enjoyment of clinical practice | 31 | 7.0% | 55 | 12.3% | 360 | 80.7% |
| Stress levels of healthcare providers | 54 | 12.1% | 58 | 13.0% | 335 | 74.9% |
| Healthcare providers self-image | 45 | 10.1% | 61 | 13.7% | 339 | 76.2% |
| Humaneness of clinical practice | 44 | 9.8% | 72 | 16.1% | 331 | 74.0% |
| The rapport between healthcare providers and patients | 41 | 9.2% | 62 | 13.9% | 342 | 76.9% |
| Personal and professional privacy | 25 | 5.6% | 40 | 8.9% | 382 | 85.5% |
| Healthcare providers access to up to-date information | 10 | 2.2% | 30 | 6.7% | 407 | 91.1% |
| Patients satisfaction with the quality of care they receive | 9 | 2.0% | 46 | 10.4% | 389 | 87.6% |
| Generalists ability to manage more complex problems | 10 | 2.2% | 47 | 10.6% | 388 | 87.2% |
| Comprehensiveness of patient care | 11 | 2.5% | 29 | 6.5% | 407 | 91.1% |
| Efficiency of clinical practice | 10 | 2.2% | 24 | 5.4% | 413 | 92.4% |
| Avoiding errors (such as overlooking drug interactions) | 19 | 4.3% | 30 | 6.7% | 398 | 89.0% |
| **EHR FUNCTIONALITY** | **Difficult** | | **No Diff** | | **Easy** | |
| Obtain and review patient information and data | 7 | 1.6% | 7 | 1.6% | 428 | 96.8% |
| Document care for my patients | 7 | 1.6% | 2 | 0.5% | 427 | 97.9% |
| View lab results for my patients | 4 | 0.9% | 4 | 0.9% | 428 | 98.2% |
| View radiology results for my patients | 4 | 0.9% | 7 | 1.6% | 423 | 97.5% |
| Prevent adverse events (e.g., drug-drug interaction, drug-allergy interaction) | 8 | 1.9% | 10 | 2.3% | 412 | 95.8% |
| Track preventive care for my patients | 6 | 1.4% | 19 | 4.4% | 407 | 94.2% |
| Manage chronic disease conditions for my patients | 4 | 0.9% | 20 | 4.6% | 409 | 94.5% |
| Manage orders | 9 | 2.1% | 6 | 1.4% | 419 | 96.5% |
| Manage referrals | 7 | 1.6% | 9 | 2.1% | 416 | 96.3% |
| Provide patient educational materials | 8 | 1.9% | 26 | 6.1% | 393 | 92.0% |
| Analyze outcomes of care | 7 | 1.6% | 13 | 3.0% | 410 | 95.3% |
| Access the EHR from offsite locations | 6 | 1.4% | 3 | 0.7% | 418 | 97.9% |
| Communicate with my colleagues to coordinate care | 10 | 2.3% | 21 | 4.8% | 411 | 93.0% |
| Communicate with my patients | 10 | 2.4% | 26 | 6.2% | 385 | 91.4% |
| Enhance the continuity of care the hospital can provide | 7 | 1.6% | 11 | 2.6% | 410 | 95.8% |

experience (87.7%). Respondents stated that they would recommend the EHR to similar practices (81.9%). In terms of negative views of the EHR, most respondents did not believe it interfered with their work (84.4%), would not want to cease use of the EHR (86.6%) and did not believe that the EHR added more work to their normal workload (82.3%). Table 5 summarises the overall evaluation of the EHR.

## Differences among healthcare workers

Regarding healthcare categories, there was significant difference among healthcare categories with prior use of EHR and if the EHR had changed their work as compared to paper based. More doctors as compared to nurses or allied staff reported using prior EHR systems (35.5% versus 13.7% versus 23.0%; p = 0.004). Nurses had reported at 95.4% for EHR to be much better as compared to paper-records as compared to doctors at 91.1% and allied staff at 78.9% (p = 0.004).

**Table 5. EHR Overall Evaluation of study participants.**

| OVERALL EVALUATION OF THE EHR | Disagree | | NOR | | Agree | |
|---|---|---|---|---|---|---|
| To me, use of this EHR is easy | 6 | 1.3% | 22 | 4.9% | 423 | 93.8% |
| The EHR screens are user friendly | 13 | 2.9% | 22 | 4.8% | 419 | 92.3% |
| This EHR provides all functionalities that I expect | 14 | 3.1% | 32 | 7.0% | 408 | 89.9% |
| Overall, I am satisfied with my experience with this EHR | 11 | 2.4% | 45 | 9.8% | 401 | 87.7% |
| I would recommend this EHR to other similar practices | 15 | 3.3% | 67 | 14.8% | 372 | 81.9% |
| My colleagues have negative opinions about this EHR | 271 | 59.3% | 133 | 29.1% | 53 | 11.6% |
| Use of this EHR interferes with my work | 385 | 84.4% | 25 | 5.5% | 46 | 10.1% |
| I would be in favor of ceasing use of this EHR in our practice | 394 | 86.6% | 14 | 3.1% | 47 | 10.3% |
| Use of this EHR requires me to do more work compared to what I used to do | 377 | 82.3% | 22 | 4.8% | 59 | 12.9% |

Nurses reported higher agreement rates as compared to doctors and allied staff especially on adequate training (98.6% vs 93.5% vs 93.0%; p = 0.095) and satisfaction with technical support (95.0% vs 90.2% vs 78.4%; p = 0.019). Furthermore, on easiness of EHR functionality, nurses reported higher rates as compared to doctors and allied staff on analyzing outcomes of care (100.0% vs 87.0% vs 94.7%; p = 0.017), providing patient educational materials (97.7% vs 80.0% vs 84.7%; p = 0.017) and managing referrals (99.1% vs 91.5% vs 95.8%; p = 0.068). Nurses reported higher agreement rates as compared to doctors and allied staff, especially on overall satisfaction (92.7% vs 90.8% vs 75.4%; p = 0.009) and recommendation (86.2% vs 85.7% vs 69.9%; p = 0.009). All differences among of EHR user experiences between doctors, nurses and allied staff are presented in Supplementary tables as S1 Table.

## Discussion

Overall, the EHR deployment in the AKU Nairobi health system proved highly successful and is seen by end users as a valuable tool for patient care and coordinating activities. The high rate of fidelity, as seen with the > 90% responding positively to questions regarding EHR functionality and ease of use, demonstrates the viability of large scale EHR deployments using specific strategies for EHR implementation.

In our study, a significant proportion of the participants were young health professionals (less than 40 years old). Research conducted in Kuwait suggests that younger health professionals demonstrate greater readiness for EHR systems [17]. This trend may be attributed to younger individuals' intrinsic motivation, interest and willingness to embrace new technological advancements compared to older generations. Similarly, the high acceptance and adoption rates of EHR in this study can likely be explained by the large number of younger, dynamic and computer savvy participants [17].

There were some key differences in the responses of physicians versus other clinical staff regarding the EHR's impact on clinical care. Physicians, compared to other end users, responded less positively regarding whether the EHR improved various aspects of clinical care. Their negative responses were most pronounced when asked questions about the EHR's effect on productivity and administrative burden such as worker autonomy, patient-provider interactions, and stress on health care workers.

Despite recognizing certain limitations, physicians viewed the EHR positively and indicated they would recommend the EHR to other clinicians. These findings suggest that although the EHR system was seen as reducing productivity and increasing administrative tasks, these drawbacks were not significant enough to justify reverting back to a paper-based system. Physicians' responses in this study closely mirror those from HICs during the implementation of their EHRs, particularly the US, where productivity loss and administration burden have also been identified as major challenges during EHR implementation [18]. These factors are also amongst the main contributors to burnout amongst US physicians [19], raising concerns about potential parallels in LMICs. Further research is needed to understand how EHRs can be modified to address physicians' concerns and ensure physicians in LMICs do not experience the same levels of burnout as their

HIC counterparts. To mitigate dissatisfaction related to productivity and administrative workload, several strategies can be employed. Implementing team-based care models such as incorporating medical scribes or care coordinators can help redistribute non-clinical tasks, allowing physicians to concentrate more on patient care. Improving EHR usability through user-centered design, voice recognition, and automation tools can also reduce documentation time and enhance efficiency. Additionally, leveraging artificial intelligence to automate repetitive tasks, coupled with ongoing training and technical support, can boost physician confidence and proficiency in using these systems. Promoting work-life balance by limiting after-hours documentation and offering flexible scheduling or telemedicine options may also contribute to improved job satisfaction. Finally, fostering a supportive organizational culture through leadership engagement, wellness programs, and regular feedback mechanisms can play a critical role in reducing burnout and enhancing physician well-being.

Slow system speed and unplanned downtimes have been shown to negatively impact the end-user experience [20,21]. Similarly, when the EHR was unresponsive, many respondents felt the backup system was inadequate or simply not present. Nurses in particular were acutely affected and felt they were unable to achieve their basic clinical tasks when the system was not functioning, compared to physicians and allied health staff. Our findings, although unique, could be attributed to the constant use of EHRs by the nursing team at the patient's bedside, compared to physician and allied health care staff. This study has helped us develop better protocols and education methods to ensure that all clinical providers are aware of the back-up process when the EHR system is down.

Previous EHR implementation projects demonstrated varying degrees of success and can be used as comparisons to our study. Uwambaye et al found similar positive responses regarding user satisfaction and data quality in their survey of end users in a central tertiary hospital in Rwanda [22]. They were, however, limited to the central referral hospital, compared to the 50 outreach centers additionally surveyed in our study. EHR deployments in Ethiopian Central hospitals and a network of HIV clinics in Rwanda faced the burden of dual reporting in both paper-based systems and the EHR during the first few months of deployment [8,23]. This additional administrative burden reduced the overall experience for end users. Our institution's use of a "Go Live" designation and full transition eliminated the need for dual reporting, thereby improving end users' use environment.

Procurement or enhancement of hardware has been shown to be a key factor in a successful implementation process [20,21,24,25]. Moreover, technical limitations to both hardware and software are the most frequently mentioned barriers to an implementation process. Lack of space and infrastructure have also been identified as barriers to the implementation process [26]. A majority of our end-users expressed satisfaction with the increase in hardware, especially computer terminals, both desktops and computers on wheels (COWs) that improved access to patient labs, imaging and documentation. Prior to implementation, much work was put in to ensure that adequate space was available in the clinical areas to accommodate the hardware as well as providers taking care of patients.

Furthermore, adequate training and support remains crucial for the implementation process [21,24,25]. At our institution, the use of an in-house clinical informatics team was a novel and cost-effective idea created during the planning of the EHR rollout and a vast majority of the end-users felt that training and post Go-Live support were adequate. Training local clinicians to become adept with the system and to act as a bridge with the clinical staff showed to be highly effective in training end-users and addressing technical difficulties. Their presence also demonstrated institutional backing of the EHR, a key factor in ensuring the deployment's success. Other EHR deployments mainly relied on technical staff who, while may be experienced in clinical settings, lack the context and perspectives of clinical staff. Further research on the experiences of the dedicated clinical staff may be useful for future deployments in other clinical sites.

In addition, it has been shown that successful implementations of EHR systems require the involvement of administrative personnel and IT specialists who understand the back end of the system and offer vital technical support to end users. At our institution, the readily available technical support (electronically and physically) and the end-user user satisfaction with this support had a substantial contribution to the acceptance and successful implementation and integration of HER [27].

The study was done through convenience sampling of end users. The reported views may not represent the general views of EHR end-users. In addition, respondents with stronger views, whether positive or negative, may be more likely to respond, compared to purposeful sampling. Participants were also surveyed a year after the deployment of the EHR. Their views on the impact and ease of use may have changed as they became more accustomed to the EHR and understood the ramifications of the new system. While our current study focused on survey-based assessment of end-user experiences with an EHR platform, we recognize that emerging computational methods—using machine learning models—can offer valuable insights when applied to relevant healthcare datasets and implementation studies. Integrating such approaches in future studies may enhance understanding of user behavior patterns, predict adoption trends, or optimize support mechanisms in EHR rollouts.

## Conclusion

Our study demonstrates that health care workers felt the introduction of an EHR was effective in improving clinical care for AKU patients. The survey also showed that strategies deployed by AKU were effective in ensuring EHR utilization and acceptance amongst HCWs. They had favorable views on the EHR use environment, impact and functionality, views which were shared across job responsibilities, though physicians did share hesitancy regarding the EHR's impact on their work. Overall, these strategies can be looked to as guidelines for other LMIC health systems deploying similar EHR systems.

## Supporting information

**S1 Table. Differences of EHR experiences among different health care categories.**
(DOCX)

## Acknowledgments

We would like to thank the Aga Khan University Hospital, Nairobi staff for their cooperation in this study. We would also like to thank Annastacia Mbithi and the Department of Medicine for sending reminders to the staff for participation in the study.

## Author contributions

**Conceptualization:** Jasmit Shah, Zohray Talib, Sayed K. Ali.

**Data curation:** Jasmit Shah, Zamanali Khakhar, Armando Ruiz.

**Formal analysis:** Anmol Shrestha, Jasmit Shah, Zohray Talib.

**Investigation:** Anmol Shrestha, Sayed K. Ali.

**Methodology:** Jasmit Shah, Zamanali Khakhar, Sayed K. Ali, Armando Ruiz.

**Supervision:** Zohray Talib, Sayed K. Ali.

**Validation:** Anmol Shrestha, Zamanali Khakhar.

**Writing – original draft:** Anmol Shrestha, Jasmit Shah.

**Writing – review & editing:** Jasmit Shah, Zamanali Khakhar, Zohray Talib, Sayed K. Ali, Armando Ruiz.

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
