## [Decision Letter · Decision Letter 0]

30 Jul 2025

PONE-D-25-30375END USER EXPERIENCES OF AN ELECTRONIC HEALTH RECORDS PLATFORM IN A TERTIARY HOSPITAL SYSTEM IN KENYAPLOS ONE

Dear Dr. ali,

Thank you for submitting your manuscript to PLOS ONE. After careful consideration, we feel that it has merit but does not fully meet PLOS ONE’s publication criteria as it currently stands. Therefore, we invite you to submit a revised version of the manuscript that addresses the points raised during the review process.

We look forward to receiving your revised manuscript.

Kind regards,

Hamufare Dumisani Mugauri, Ph.D. Medicine and Health Sciences

Academic Editor

PLOS ONE

Journal Requirements: 

2. Please ensure that you have specified a) Did participants provide their written or verbal informed consent to participate in this study?

b) If consent was verbal, please explain i) why written consent was not obtained, ii) how you documented participant consent, and iii) whether the ethics committees/IRB approved this consent procedure."

- In consent please state in Ethics Method section and manuscript if it is written or verbal. If consent was verbal, please explain a) why written consent was not obtained, b) how you documented participant consent, and c) whether the ethics committees/IRB approved this consent procedure.

3. Please include a complete copy of PLOS’ questionnaire on inclusivity in global research in your revised manuscript. Our policy for research in this area aims to improve transparency in the reporting of research performed outside of researchers’ own country or community. The policy applies to researchers who have travelled to a different country to conduct research, research with Indigenous populations or their lands, and research on cultural artefacts. The questionnaire can also be requested at the journal’s discretion for any other submissions, even if these conditions are not met.  Please find more information on the policy and a link to download a blank copy of the questionnaire here: https://journals.plos.org/plosone/s/best-practices-in-research-reporting. Please upload a completed version of your questionnaire as Supporting Information when you resubmit your manuscript.

Reviewers' comments:

Reviewer's Responses to Questions

**Comments to the Author**

1. Is the manuscript technically sound, and do the data support the conclusions?

Reviewer #1: Yes

Reviewer #2: Yes

2. Has the statistical analysis been performed appropriately and rigorously? 

Reviewer #1: Yes

Reviewer #2: Yes

3. Have the authors made all data underlying the findings in their manuscript fully available?

Reviewer #1: Yes

Reviewer #2: Yes

4. Is the manuscript presented in an intelligible fashion and written in standard English?

Reviewer #1: Yes

Reviewer #2: No

5. Review Comments to the Author

Reviewer #1: In this submission to PLOS One, the authors conducted a cross-sectional study between October 2023 and January 2024 at the Aga Khan University Hospital, Nairobi with staff involved in using the EHR system. A standardized electronic questionnaire by the authors was shared with the users and responses were captured on REDCap after obtaining consent. The majority of end users believed the EHR to be effective and appropriate to use within this specific and unique healthcare system. The authors note that specific strategies deployed by institutions were also successful in ensuring high rates of EHR usage and can be looked at as a blueprint for future EHR deployments in other sub-Saharan African healthcare systems.

I find this manuscript to be of interest to healthcare researchers as well as readers of this journal. As such, I am generally supportive of publication with a few minor but essential comments. While the authors use conventional surveys, there have been prior studies using machine learning for healthcare, which should be noted: Environ. Sci. Technol. Lett. 2023, 10, 1017–1022 and Environ. Sci. Technol. 2019, 53, 13970–13980. Specifically, these prior studies have shown that advanced machine learning approaches can provide more details into the underlying mechanisms of biological activity that can complement the experiments. I am not asking the authors to carry out new machine learning calculations at all, but these prior treatments should be noted since these machine learning techniques are well known and a mature field now.

Reviewer #2: Thank you for the opportunity to review this important and timely study, which examines the perceptions and satisfaction of end-users with a newly implemented electronic health record (EHR) system in a tertiary hospital system in Kenya. This work has the potential to inform EHR implementation strategies in different hospital systems and other countries with similar environment.

Overall Comments

• This study addresses a critical area and may serve as a valuable guide for implementing EHR systems in hospital settings, especially in regions with comparable infrastructures and constraints.

• However, to enhance its relevance and applicability, the manuscript should include more detailed descriptions of how the EHR system was implemented in their hospital system. Please refer to my specific suggestions in the Introduction section.

Introduction

• Please spell out the abbreviation "ICT" on first use.

• Include more details about the EHR implementation strategy. For instance:

o What types of experts were included on the ICT team?

o What were the specific roles of clinical informatics team members?

o How frequently were town halls held, in which format (online or offline), and what topics or updates were covered during these sessions?

Methods

• Please provide a reference for the Michigan Public Health Institute EHR End-User Survey.

• A large number of statistical tests were conducted, which raises the risk of Type I error. Please apply an appropriate correction method (e.g., Bonferroni or Benjamini-Hochberg) to control for multiple comparisons.

• The rationale for comparing outcomes by gender is unclear, and these findings were not discussed. I recommend removing this comparison and focusing on comparisons across different professions.

Results

• Please provide summary tables for EHR impact, functionality, and overall evaluation within the main text.

• Clearly reference any supplementary tables when discussing results in the text.

• For supplementary materials, consider separating topics into individual tables (e.g., one table for experience, another for environment, etc.).

• Review number rounding throughout the manuscript. For example, 78.99% should be rounded to 79.0%, not 78.9%.

Discussion

• Please include potential solutions or recommendations to address physicians’ concerns—particularly regarding lower satisfaction related to productivity and administrative burden.

6. PLOS authors have the option to publish the peer review history of their article (what does this mean? ). If published, this will include your full peer review and any attached files.

**Do you want your identity to be public for this peer review?** For information about this choice, including consent withdrawal, please see our Privacy Policy .

Reviewer #1: No

Reviewer #2: No

---

## [Author Response · Author response to Decision Letter 1]

8 Aug 2025

END USER EXPERIENCES OF AN ELECTRONIC HEALTH RECORDS PLATFORM IN A TERTIARY HOSPITAL SYSTEM IN KENYAPLOS ONE

[PONE-D-25-30375]

We thank the reviewers for pointing out the comments below and we have tried to address the comments and strengthen the manuscript.

Response to Comments

Overall Editor Comments

Response:

1. Please ensure that you have specified a) Did participants provide their written or verbal informed consent to participate in this study? b) If consent was verbal, please explain i) why written consent was not obtained, ii) how you documented participant consent, and iii) whether the ethics committees/IRB approved this consent procedure."

Response: Participants provided written informed consent electronically prior to their enrollment in the study. This consent procedure, including the use of electronic consent, was reviewed and approved by the Aga Khan University, Institutional Scientific Ethics and Review Committee (AKU-ISERC).

2. In consent please state in Ethics Method section and manuscript if it is written or verbal. If consent was verbal, please explain a) why written consent was not obtained, b) how you documented participant consent, and c) whether the ethics committees/IRB approved this consent procedure.

Response: Participants provided written informed consent electronically prior to their enrollment in the study. This consent procedure, including the use of electronic consent, was reviewed and approved by the Aga Khan University, Institutional Scientific Ethics and Review Committee (AKU-ISERC).

3. Please include a complete copy of PLOS’ questionnaire on inclusivity in global research in your revised manuscript. Our policy for research in this area aims to improve transparency in the reporting of research performed outside of researchers’ own country or community. The policy applies to researchers who have travelled to a different country to conduct research, research with Indigenous populations or their lands, and research on cultural artefacts.

Response: We have included a copy of the PLOS’ questionnaire on inclusivity in global research.

Response: We have deleted the Ethics statement in declarations. It only now appears in the methods.

5. Please include captions for your Supporting Information files at the end of your manuscript, and update any in-text citations to match accordingly. Please see our Supporting Information guidelines for more information:

http://journals.plos.org/plosone/s/supporting-information.

Response: We have included captions at the end of the manuscript.

Reviewer #1:

6. As such, I am generally supportive of publication with a few minor but essential comments. While the authors use conventional surveys, there have been prior studies using machine learning for healthcare, which should be noted: Environ. Sci. Technol. Lett. 2023, 10, 1017–1022 and Environ. Sci. Technol. 2019, 53, 13970–13980. Specifically, these prior studies have shown that advanced machine learning approaches can provide more details into the underlying mechanisms of biological activity that can complement the experiments. I am not asking the authors to carry out new machine learning calculations at all, but these prior treatments should be noted since these machine learning techniques are well known and a mature field now.

Response: We thank the reviewer for their insightful comment and for highlighting the referenced studies on the use of machine learning (Environ. Sci. Technol. Lett. 2023, 10, 1017–1022 and Environ. Sci. Technol. 2019, 53, 13970–13980). We agree that machine learning is a powerful and mature tool in biomedical research and acknowledge the value it brings in elucidating biological mechanisms. However, we note that the focus of our current study is distinct, as it centers on assessing end-user experiences of an electronic health records platform in a tertiary hospital setting in Kenya. Our methodological approach, which is based on conventional surveys, is designed to capture usability, satisfaction, and contextual factors influencing EHR adoption and interaction from the perspective of frontline healthcare providers.

While machine learning may offer valuable insights in other biomedical domains, particularly in bioactivity prediction and systems biology, it falls outside the immediate scope and objectives of our study. We appreciate the broader context raised by the reviewer and have included a brief note in the discussion to acknowledge the growing role of advanced analytical techniques, including machine learning, in complementary areas of health systems research.

Reviewer #2:

This study addresses a critical area and may serve as a valuable guide for implementing EHR systems in hospital settings, especially in regions with comparable infrastructures and constraints.

Introduction

7. Please spell out the abbreviation "ICT" on first use. Include more details about the EHR implementation strategy. For instance: What types of experts were included on the ICT team? What were the specific roles of clinical informatics team members? How frequently were town halls held, in which format (online or offline), and what topics or updates were covered during these sessions?

Response: We have revised the introduction and included more on the implementation strategy as suggested in the comments.

Methods

8. Please provide a reference for the Michigan Public Health Institute EHR End-User Survey.

Response: We have included the reference for the survey.

9. A large number of statistical tests were conducted, which raises the risk of Type I error. Please apply an appropriate correction method (e.g., Bonferroni or Benjamini-Hochberg) to control for multiple comparisons.

Response: We have added the adjusted p values using Bonferroni correction.

10. The rationale for comparing outcomes by gender is unclear, and these findings were not discussed. I recommend removing this comparison and focusing on comparisons across different professions.

Response: We have removed the table on gender and revised accordingly.

Results

11. Please provide summary tables for EHR impact, functionality, and overall evaluation within the main text.

Clearly reference any supplementary tables when discussing results in the text.

For supplementary materials, consider separating topics into individual tables (e.g., one table for experience, another for environment, etc.).

Response: We have revised the results and added the summary tables in the main manuscript.

12. Review number rounding throughout the manuscript. For example, 78.99% should be rounded to 79.0%, not 78.9%.

Response: We have reviewed all tables and, in the percentages, we have rounded to one decimal point.

Discussion

13. Please include potential solutions or recommendations to address physicians’ concerns—particularly regarding lower satisfaction related to productivity and administrative burden.

Response: We have provided some recommendations in the discussion.

---

## [Editor Report · Decision Letter 1]

28 Aug 2025

END USER EXPERIENCES OF AN ELECTRONIC HEALTH RECORDS PLATFORM IN A TERTIARY HOSPITAL SYSTEM IN KENYA

PONE-D-25-30375R1

Dear Dr. Sayed,

We’re pleased to inform you that your manuscript has been judged scientifically suitable for publication and will be formally accepted for publication once it meets all outstanding technical requirements.

Kind regards,

Hamufare Dumisani Mugauri, Ph.D. Medicine and Health Sciences

Academic Editor

PLOS ONE
---

## [Editor Report · Acceptance letter]

PONE-D-25-30375R1

PLOS ONE

Dear Dr. ali,

I'm pleased to inform you that your manuscript has been deemed suitable for publication in PLOS ONE. Congratulations! Your manuscript is now being handed over to our production team.

Kind regards,

on behalf of

Dr Hamufare Dumisani Mugauri

Academic Editor

PLOS ONE